# Absorbable Haemostatic Artefacts as a Diagnostic Challenge in Postoperative Follow-Up After Oncological Resection in Head and Neck Tumours: Systematic Review of Current Literature and Two Case Reports

**DOI:** 10.3390/diagnostics14232667

**Published:** 2024-11-27

**Authors:** Giorgio Barbera, Guido Lobbia, Federica Ghiozzi, Alessandra Rovescala, Carlotta Franzina, Sokol Sina, Riccardo Nocini

**Affiliations:** 1Head and Neck Department, Azienda Ospedaliera Universitaria Integrata di Verona, Piazzale Aristide Stefani 1, 37126 Verona, Italy; giorgio.barbera@aovr.veneto.it (G.B.); riccardo.nocini@aovr.veneto.it (R.N.); 2Unit of Maxillofacial Surgery, Head and Neck Department, University of Verona, Piazzale Ludovico Antonio Scuro 10, 37134 Verona, Italy; federica.ghiozzi@studenti.univr.it (F.G.); alessandra.rovescala@studenti.univr.it (A.R.); 3Unit of in Pathology, University of Verona, Piazzale Ludovico Antonio Scuro 10, 37134 Verona, Italy; 4Department of Diagnostics and Public Health, Section of Pathology, Azienda Ospedaliera Universitaria Integrata di Verona, Piazzale Aristide Stefani 1, 37126 Verona, Italy; sokol.sina@aovr.veneto.it

**Keywords:** oxidised resorbable cellulose, head and neck cancer, pseudotumour, tumour recurrence, haemostatic agents

## Abstract

***Objectives*:** This article aims to define the clinical, radiological, and pathological characteristics of non-resorbed oxidised cellulose-induced pseudotumours to raise awareness among surgeons and radiologists, to prevent misdiagnosis, and avoid unnecessary invasive procedures and delays in adjuvant oncological treatments. ***Methods*:** A systematic review of oxidised resorbable cellulose (ORC)-induced pseudotumours of the head and neck was conducted following PRISMA 2020 guidelines. Articles were retrieved from PubMed, Scopus, Cochrane, and Web of Science. Two ORC-induced pseudotumour cases from the Maxillofacial Surgery Department of Verona are also presented. ***Results*:** In most cases, pseudotumours were monitored using ultrasound. Further investigations included CT, MRI, PET-CT, and scintigraphy. Ultrasound images showed stable, elongated, and non-homogeneous masses. In CT scans, pseudotumours showed a liquefied core, and none or only peripheral enhancement. In MRI, pseudotumours presented none or only peripheral enhancement, and a heterogeneous pattern in T2-weighted images. 18-FDG PET scans demonstrated an FDG-avid mass (SUV 7.5). Scintigraphy was inconclusive. Cytology indicated a granulomatous reaction without neoplastic cells. Where surgical excision was performed, a granulomatous reaction with the presence of oxidised cellulose fibres was confirmed. ***Conclusions*:** Surgeons should consider artifacts from retained oxidised absorbable haemostatic material when suspecting tumour recurrence or metastasis on postoperative imaging, especially if certain features are present. Fine-needle aspiration cytology (FNAC) is a useful diagnostic tool, but surgical excision may be needed if FNAC is inconclusive or impractical. Collaboration between surgeons and radiologists is essential to avoid misdiagnosis and delays in treatment. Documenting the use and location of haemostatic material in operative reports would aid future understanding of these phenomena.

## 1. Introduction

The use of haemostatic agents is crucial for preventing significant blood loss both in elective and emergency surgical procedures. Oxidised resorbable cellulose (ORC) is a highly effective biodegradable and biocompatible haemostatic. In the last 50 years, it has emerged as one of the most used haemostatic agents in surgical practices due to its ease of use, favourable biocompatibility, bactericidal activity, and bio-absorption characteristics [1,2].

There are several types of haemostatic materials, which can be divided into patch type (including gauze, nonwoven fabric, filaments, sponges, and membranes), particle powder type (including mineral haemostatic particles and haemostatic powders), fluid sealing type (including biological glue, chemical glue, wax, and liquid foam), and tactile expansion type (including compressed cotton and solid glue). The choice of the haemostatic material largely depends on its characteristics and suitability to the wound type [3].

Absorbable haemostatics include animal, plant, and synthetic derivatives [4]. Oxidised regenerated cellulose (ORC) is a cellulose derivative, obtained from wood pulp, and it is considered one of the most commonly used local surgical haemostatic biomaterials. The dual process of oxidation and regeneration applied to the alpha-cellulose results in the formation of a uniform chemical–physical molecular structure called rayon, characterised by fibres of uniform diameter and good tensile strength. Furthermore, the controlled and uniform oxidation process endows the agent with several defining characteristics, including its haemostatic action, absorbability, and antimicrobial properties. It was demonstrated that all formats of ORC compounds are well tolerated and undergo substantial absorption within 14 days from implantation. However, it has been observed that the process can take up to 8 weeks to be completed [5].

The patch-type ORC compound is the most used haemostatic material at our Institution. It consists of several layers that may be utilised separately. Upon contact with blood, each layer becomes transparent, allowing the operator to verify the effectiveness of haemostasis. Although the ORC compounds’ haemostasis mechanisms have not been fully understood, their haemostatic action seems to be based on both physical and chemical properties. On the physical side, the material works by capturing coagulation factors, platelets, red blood cells, and other components, thereby accelerating blood clotting. At the same time, chemical properties lower the pH and convert haemoglobin to acidic haematin, processes that further accelerate clot formation [6].

Despite their widespread use, ORC compounds can be associated with postoperative complications particularly concerning their biocompatibility and reabsorption. The literature contains several examples of foreign body reactions to retained ORC. These can remain completely asymptomatic or can induce mass effect symptoms. Retained ORC is often detected in radiological exams and can be misinterpreted by radiologists as a concerning finding. This especially happens after surgical intervention in oncological patients. Retained ORC may appear as a mass-mimicking abscess, haematoma, granuloma, tumour recurrence, or tumour persistence, therefore posing a significant diagnostic challenge [1].

A systematic review of the literature regarding masses induced by retained ORC in the head and neck district is conducted in this article. Two cases of pseudotumours induced by retained ORC observed in our department are also presented and compared to the literature.

The objective of this study is to identify the radiological and histopathological characteristics of lesions caused by the non-absorption of surgical haemostatic material in the head and neck district, to raise awareness of these findings among surgeons and radiologists, to prevent misdiagnoses or unnecessary invasive interventions, and to avoid delaying any necessary adjuvant oncological treatments.

## 2. Case Presentation

Case reports were written according to SCARE guidelines [7].

### 2.1. Case 1

A 69-year-old Caucasian male was referred to the outpatient section of the Maxillofacial Surgery Unit of the University of Verona in September 2023 for an ulcerated neoformation of the left hemi-mandibula mucosae. The lesion was immediately biopsied, revealing a squamous cell carcinoma and a whole-body CT scan with contrast enhancement (CECT) was performed. The tumour stage at the time of diagnosis was classified as cT4 N0 M0. In October 2023, the patient underwent a left hemi mandibulectomy (B2 deficit according to Della Monaca and Valentini classification [8]) with ipsilateral neck dissection and scapular and latissimus dorsi flap reconstruction [9]. The operation was carried out successfully and the postoperative course was uneventful. The histopathological examination yielded a diagnosis of squamous cell carcinoma, pT4a G2 pN0 cM0, with no evidence of adverse features, indicating a stage IVa tumour.

A whole-body CT scan with contrast medium was conducted one month following surgery and unexpectedly revealed the presence of a formation in the left posterior jugular area (Figure 1). The radiologist described the formation as a lymph node with a major axis of 65 mm and with areas of colliquation.

Given the suspicion of metastasis to the lymph nodes, 18-FDG PET/CT and an echo-guided fine-needle aspiration (FNAC) were subsequently conducted. The PET/CT scan revealed the presence of an ovoid mass with elevated 18-FDG uptake (SUV 7.5) at the aforementioned location (Figure 2).

The sonography described an elongated, non-homogeneous hyperechogenic formation with an expansive growth pattern. The cytological examination revealed the presence of necrotic debris, neutrophilic granulocytes, and rare multinucleated histiocytic giant cells. However, no atypical cells were identified (Figure 3).

Given the 18-PET-CT result, and despite the negative result of the cytological examination, the multidisciplinary team proposed a radical neck dissection to the patient, who immediately refused to undergo another surgery. Moreover, even if radiotherapy was an option, it could not be performed without a definitive histological diagnosis of the mass. CT scans and FNAC were then repeated at three months post-operatively: the CT images and the sonographic findings were consistent with the previous ones; it is noteworthy that the second cytological examination revealed the presence of birefringent material, with macrophages, and isolated foreign body-like giant multinucleated cells and again no atypical cells. Considering these latest findings, and the surgeon’s assertion that the formation was located where a considerable quantity of Tabotamp^®^ (Ethicon, a Johnson and Johnson company, Somerville, MA, USA) had been left during surgery, a foreign body reaction to ORC was prompted as the most plausible diagnosis.

A rigorous follow-up program was planned. CT at six months post-surgery demonstrated unaltered images. Nine months following surgery, the patient underwent magnetic resonance imaging (MRI) and fine-needle aspiration cytology (FNAC). During MRI imaging, a persistent left mass was observed, with no evidence of a restricted signal in DWI (diffusion-weighted imaging) or post-contrastographic enhancement (Figure 4).

At US, the nodule was noted to be stable in dimension, and the presence of bubbles was identified within the core. Cytologic findings remained unchanged. At the subsequent follow-up visit, the patient exhibited no clinical indications of recurrence at the local site or in the lymph nodes. Based on the findings from both clinical and radiological evaluations, the patient was deemed to be free from disease at ten months’ follow-up.

### 2.2. Case 2

A 78-year-old female patient presented at the outpatient section of the Maxillofacial Surgery Unit of the University of Verona in August 2023 due to the detection of a pterygomaxillary mass of the left maxilla during magnetic resonance imaging (MRI). More than 10 years before, in 2011, the patient had undergone surgical intervention to remove an extraosseous ameloblastoma of the left upper maxilla. At the time of our consultation, the patient had already undergone ENT evaluation and a biopsy of the aforementioned pterygomaxillary mass; the histological report documented the presence of fragments of epithelial neoplasia with a plexiform architecture, consisting of elements with partly basaloid and partly squamous morphology, compatible with a recurrence of ameloblastoma. Notwithstanding these procedures, swelling persisted and a CT scan was prescribed confirming the presence of a solid lesion in the left masticatory space with homogeneous density, whose measurements were 6.5 × 3.1 × 5.5 cm. The mass also infiltrated the homolateral maxillary sinus. The patient subsequently underwent an endoscopic sinonasal biopsy of the lesion [10]. The histological examination confirmed the presence of an ameloblastoma recurrence. Therefore, the patient underwent an excision of the aforementioned pterygomaxillary mass. The final histological examination confirmed the lesion to be a plexiform ameloblastoma with plexiform and basaloid architecture.

Five months later, follow-up CT and MRI scans with contrast were performed. MRI imaging revealed the presence of material characterised by a more peripheral solid portion with slight contrast enhancement and a central portion with a liquid signal, extensively occupying the left maxillary sinus, the pterygopalatine fossa, and the infratemporal fossa, reaching up to the floor of the orbit (Figure 5).

CT imaging showed residual tissue with a peripheral solid portion with contrast enhancement and a central hypodense area likely representing fluid. This lesion extensively occupied the left maxillary sinus, the pterygomaxillary fossa, and the infratemporal fossa, infiltrating the left orbit.

On suspicion of a second recurrence of ameloblastoma, one more additional surgical procedure was performed. During the procedure, a mass likely resembling haemostatic material from a previous surgery and inflammatory tissue was removed. During the procedure, a portion of the removed material was sent for intraoperative histological examination. The pathologist reported it as “histiocytic-macrophagic nature, mixed with neutrophil granulocytes and calcific-like material”.

The final histological and immunohistochemical examinations did not document any evidence of ameloblastoma localization (negativity for CK19), and instead revealed the presence of fibro-inflammatory tissue fragments with giant cell chronic inflammation enclosing exogenous material (Figure 6).

Considering the histological examination, imaging results, and intraoperative findings, the mass was classified as a probable pseudotumour due to non-resorbed haemostatic material. Eventually, the patient was considered free from disease at six months’ follow-up.

## 3. Materials and Methods

This systematic review was conducted following the PRISMA 2020 guidelines [11]. The protocol followed is outlined in Figure 7. The search strategy was developed using the PICO method, as depicted in Table 1. The search was performed across four electronic databases (PubMed (National Library of Medicine, Bethesda, MD, USA) Scopus (Elsevier, Amsterdam, The Netherlands), Cochrane (The Cochrane Collaboration, London, UK), and Web of Science (London, UK)) without time restrictions. Only English-language articles that addressed the differential diagnosis between potential disease recurrence and the presence of an inflammatory response to resorbable haemostatic material left after the surgical removal of a primary tumour, particularly in the head and neck region, were included. Exclusion criteria encompassed articles written in languages other than English, literature reviews, incomplete manuscripts, and studies reporting on disease recurrence outside the head and neck region. Neurosurgery papers were also excluded. The selected articles were subsequently read in full and summarised.

## 4. Results

A total of 108 scientific articles written in English, with no restriction on publication year, were retrieved from four databases (PubMed, Scopus, Cochrane, Web of Science). After removing duplicates and automatically excluding ineligible records, 78 papers were excluded based on keywords, title, and abstract content, and the authors thoroughly reviewed the remaining 7. Finally, all of these seven were included in the final pool. These articles were fully read and are summarised in Table 2.

Overall, the review of the seven eligible articles allowed us to collect information about 26 clinical cases. The two cases presented in this article were incorporated into the results. Therefore, a total of 28 cases were included in the review.

The mean age of patients was 53 years (range: 27 to 78). The female–male ratio was 2:1. Surgical sites were the following: thyroid (24 cases, 85.6%), nasopharynx (1 case, 3.6%), temporomandibular joint (TMJ) (1 case, 3.6%), oral cavity (1 case 3.6%), and maxillary sinus (1 case, 3.6%). Amongst patients with thyroid affection, primary surgery consisted of 7 partial thyroidectomies and 17 total thyroidectomies. Amongst patients diagnosed with tumours in other head and neck districts, primary surgery consisted of radical excision of a nasopharyngeal carcinoma; total resection of chondrosarcoma of the temporomandibular joint; hemi-mandibulectomy with homolateral neck dissection and reconstruction with free flap for squamous cell carcinoma of the retromolar trigone; total resection of ameloblastoma of the maxillary sinus.

All patients were diagnosed with suspicious masses described as tumour recurrence or tumour persistence at post-operative imaging. The average time between primary surgery and suspected mass detection was 447 days (approximately 15 months).

### 4.1. Radiological Data

In most cases, pseudotumours were identified and monitored using ultrasound. Further investigations included CT scans with contrast enhancement (CECT), MRI, PET-CT, and scintigraphy.

From the review conducted on the selected articles, an ultrasound examination was performed on 25 patients. On ultrasound, pseudotumours caused by retained oxidised resorbable cellulose typically appeared as elongated masses with an expansive growth pattern. They were usually hypo- to iso-echoic or characterised by a heterogeneous signal, lacked central flow on Doppler, and were without calcifications. Where ultrasound monitoring was performed over time, the lesions remained mostly stable; in some cases, they reduced in size, eventually disappearing entirely.

Only three patients from the selected pool underwent a CECT. In two of the three patients, the pseudotumour was characterised by the presence of a liquefied core. In one case, only peripheral signal enhancement was observed after contrast administration, in another case, the mass did not show any uptake, and in one case contrast enhancement was not specified.

MRI was performed on three patients. A well-demarcated appearance of the pseudotumour was common in these three. Peripheral enhancement was observed after contrast administration in two of them, whereas in the last one, no uptake was described. In one case, the presence of a liquefied portion was described in a T2-weighted image. In another patient, the T2-weighted image description is of a heterogeneous pattern that does not exclude the presence of a liquified portion.

Scintigraphy was performed on one patient, showing a non-uptaking formation.

One patient underwent an 18-FDG PET-CT scan, where the pseudotumour showed uptake (SUV 7.5).

### 4.2. Pathological Data

Out of the entire case series, eight patients underwent FNAC for cytological examination. No neoplastic cells were found in any patient. The biopsy material was predominantly characterised by the presence of inflammatory cells, such as granulocytes, macrophages, multinucleated giant cells, necrotic debris, and residual acellular material, which was later identified as compatible with the structure of cellulose.

Excision of the pseudotumour and subsequent histological analysis was performed in three patients. In these cases, as well, no evidence of disease was found. Histological examination in each of these patients identified a significant granulomatous reaction, within which fibres of oxidised resorbable cellulose were present.

In the entire case series, regardless of the monitoring or interventional strategy adopted, no lymph node or local disease recurrence was observed, supporting the diagnostic hypothesis of artifacts originating from the inflammatory response to retained oxidised cellulose.

Table 3 provides detailed descriptions of the radiological and pathological findings observed.

## 5. Discussion

Oxidised regenerated cellulose compounds are widely used in surgery as haemostatic materials. Despite being bioabsorbable and biocompatible, with an average absorption time ranging from 7 to 14 days, they may sometimes persist longer in the body and be responsible for foreign body inflammatory reactions. Signs of ORC retention are identifiable on radiological images and can frequently mimic tumour recurrence or disease persistence, becoming particularly alarming in oncological patients who have undergone primary tumour resection [19,20,21]. Multidisciplinary discussion between surgeons and radiologists when encountering specific findings in postoperative imaging is key to preventing misdiagnosis of these formations and possible delays in oncological treatments [22,23].

In this article, the review of the current literature concerning ORC use in the head and neck district permitted us to gather information about the radiological characteristics associated with retained ORC pseudotumours in the most frequently used radiological examinations (ultrasound—US, computed tomography—CT, magnetic resonance imaging—MRI, PET-CT, and scintigraphy), and avoid misinterpretation of these artifacts eventually.

From our review, at US examination, pseudotumours resulting from the retention of haemostatic materials are mostly described as elongated, hypo- or iso-echoic, non-homogeneous, well-defined formations that, upon Doppler examination, demonstrate no central flow. In some cases, punctate hyperechoic echoes were described. Similar findings have been observed in other surgical sites where pseudotumours from ORC have been investigated. According to Giuliani et al., in breast surgery, the most common US pattern is of complex, well-encapsulated, hypoechoic masses, and the presence of hyperechoic nodules within the formation that the authors refer to as “floating isles” [17]. Their non-homogeneous appearance on ultrasound is a common feature in the literature, and it was found also in the above-presented case 1. This phenomenon seems to depend on the specific nature of the granulomatous reaction and the uneven and progressive absorption of the retained material [12,24]. Although ultrasound is an invaluable diagnostic tool for evaluating the head and neck region, it is not always adequate for a comprehensive assessment of this area or for follow-up in oncology patients.

Despite the limited number of patients (3/28) who underwent CT with contrast enhancement, the review has identified that CT characteristics of ORC-induced pseudotumours typically include well-defined masses with a colliquated core and, in some cases, the presence of peripheral contrast enhancement.

The presence of a colliquated core has been well documented in the literature [19]. S. T. Young et al. described the appearance of abdominal pseudotumours induced by ORC as low-attenuation areas with a water-similar density and trapped gas bubbles. Similarly, Roshkovan et al. reported that the most common finding in pseudotumours induced by ORC is the presence of focal gas collections within mixed-attenuation masses, without internal air-fluid levels. Notably, this last article emphasised the importance of radiologists being aware of the use of ORC during surgical procedures, considering that differential diagnosis between ORC-retained pseudotumours and post-operative complications can be challenging [25].

Peripheral contrast enhancement is a further finding commonly presented in ORC-induced pseudotumours and is likely due to a foreign body inflammatory response or granulation tissue ingrowth, as noted by Morani et al. in their literature review [26]. This enhancement pattern has been described also by Mayank et al. [27] and by Roshkovan et al. [25], who noticed the contrast enhancement presence on CT as either central or peripheral.

In this review, only three patients underwent MRI, which revealed ORC pseudotumours as well-defined masses with hypointense signals on T1-weighted images and hyperintense or heterogeneous appearances on T2-weighted images. Two cases demonstrated peripheral enhancement following gadolinium administration.

A recent study by Giuliani et al. examined the magnetic resonance imaging (MRI) features of ORC-induced pseudotumours in a sample of 51 women who had undergone breast surgery for mammary neoplasia. The study described pseudotumours as complex masses characterised by hyperintense collections on T2-weighted images with internal hypointense nodules. It is worthy of note that all the lesions displayed peripheral enhancement in post-contrast T1-weighted images, and none of them exhibited central enhancement [28].

In magnetic resonance imaging (MRI), Carneiro et al. noticed that foreign body reactions are characterised by a hypointense halo on T1-weighted imaging, hyperintensity on fluid-sensitive sequences, and marked peripheral enhancement around the foreign body following contrast administration. These findings are comparable to those observed in ORC-induced pseudotumours. The rim enhancement, observed both on ORC induced pseudotumours and generally present in foreign body reactions, too, is likely the result of an exaggerated inflammatory response from the tissues surrounding the foreign body or the absorbable haemostatic [29].

Similar MRI characteristics have been observed in various surgical contexts, including neurosurgery, urology, orbital surgery, and abdominal surgery [27,30,31,32].

In case 1, the patient underwent 18-FDG PET CT, showing uptake enhancement at the level of the neck. This represented a diagnostic challenge because malignant recurrences typically show an elevated 18-FDG uptake. Notwithstanding, 18-FDG uptake is a common finding in chronic inflammation, too, and in this case the 18-FDG uptake was most likely due to the presence of an active inflammatory reaction developed at the level of an ORC-induced pseudotumour [33,34].

Although radiological features may suggest the diagnosis of an ORC-induced pseudotumour, a definitive pattern has yet to be established. Especially when these findings occur post-oncological surgery, it is not yet sufficient to rely solely on radiological examination to rule out recurrence or metastatic disease. Therefore, most cases require pathological investigation to confirm the diagnosis.

Fine-needle aspiration cytology (FNAC) represents a viable technique to determine the absence of neoplastic cells in pseudotumours, thus enabling the exclusion of malignancy, persistence, or recurrence. The analysis of the FNAC samples presented in this review revealed the presence of inflammatory cells (multinucleated giant cells, foamy macrophages, and histiocytes) and fibrous material, which is consistent with the chemical and physical characteristics of absorbable haemostatic agents. Atypical cells were identified in FNAC samples in none of the examined cases of the review. In the literature concerning surgical sites other than the head and neck, cytological examination is similarly employed to avoid unnecessary surgery and to confirm the hypothesis of a pseudotumour.

Kurian et al. found comparable cytological features in transbronchial needle aspiration (TBNA) performed to investigate lung lymphadenopathies after thoracic surgery for lung cancer. Cytological examination revealed a granulomatous reaction and the presence of amorphous fragments of acellular, birefringent material, which were ultimately identified as compatible with ORC [35]. In the absence of surgical intervention, subsequent clinical–radiological follow-up demonstrated that there was no evidence of disease progression, further supporting the cytological diagnosis.

If the location of the pseudotumour is unsuitable for FNAC, or if the mass becomes symptomatic, surgical excision becomes the appropriate option to provide a definitive diagnosis. The histological appearance of the pseudotumour is defined by the presence of fibrous material consistent with ORC fibres, granulomatous reaction, and no neoplastic cells.

The biocompatibility of ORC compounds is well documented; equally it has been reported that absorbable haemostatic generates an inflammatory response in order to be completely dissolved. Inflammatory responses may induce the formation of granulomas, and this phenomenon is far more probable when a significant quantity of ORC is left in a surgical bed [5,36].

## 6. Conclusions

In conclusion, head and neck surgeons should consider the possibility of artifacts from retained oxidised absorbable haemostatic material when evaluating post-operative imaging for suspected tumour recurrence, persistence, or nodal metastasis, especially in the presence of specific radiological features. Fine-needle aspiration cytology (FNAC) is a useful and minimally invasive diagnostic tool that should be employed when feasible, as the cytological features are usually sufficiently clear to provide a definitive diagnosis. Surgical excision should be employed when the anatomical location precludes the use of cytology, when the cytological findings are inconclusive and fail to exclude the possibility of recurrent disease or tumour persistence, or when the mass is clinically symptomatic. Surgeons and radiologists must collaborate when encountering specific findings in postoperative imaging to prevent misdiagnosis of these formations and subsequent delays in oncological treatments. Additionally, surgeons should document the use and location of haemostatic material used during surgery in their surgical reports to facilitate the assessment of retained ORC-induced pseudotumours.

## Figures and Tables

**Figure 1 diagnostics-14-02667-f001:**
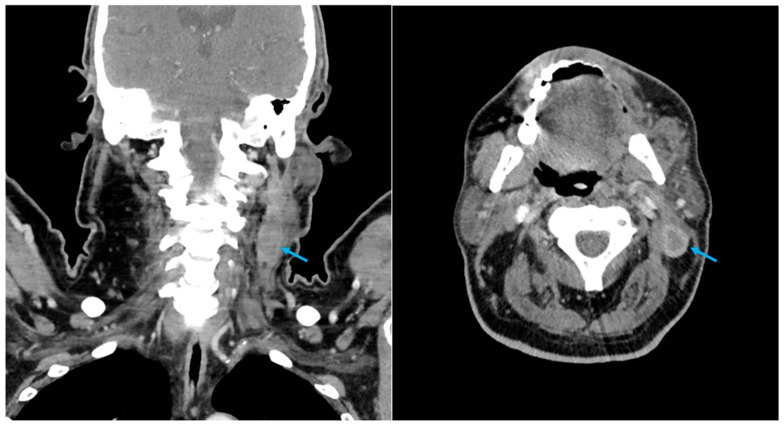
Case 1 CT scan with CE, coronal and axial view. Retained ORC pseudotumour is indicated by the arrow.

**Figure 2 diagnostics-14-02667-f002:**
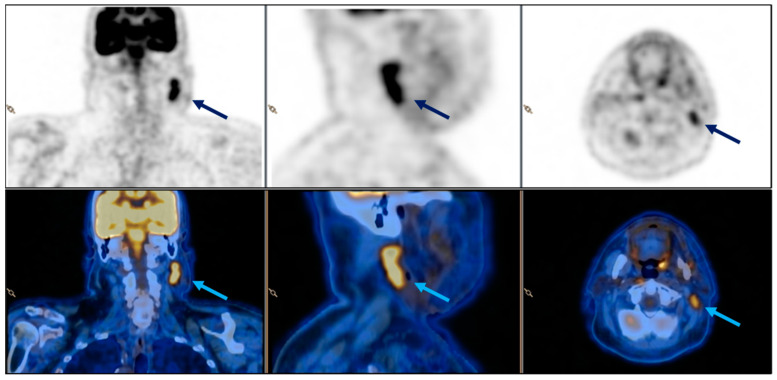
Case 1, 18-FDG PET CT, coronal, sagittal, and axial view. Retained ORC pseudotumour is indicated by the arrow.

**Figure 3 diagnostics-14-02667-f003:**
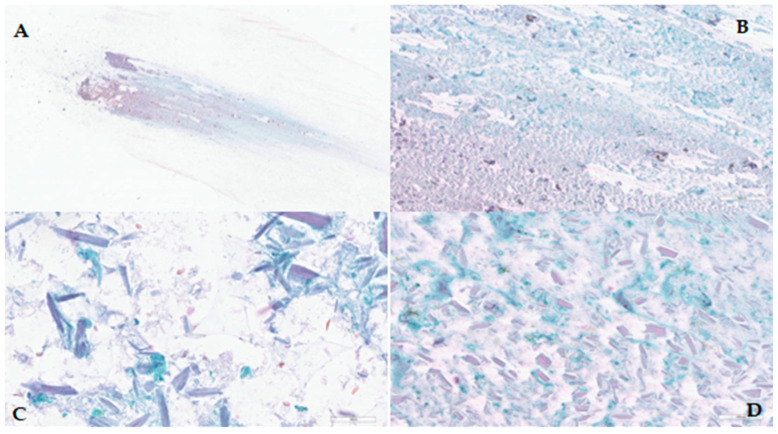
Case 1: (**A**) Cytologic slide, Papanicolaou stain, (**B**) Abundant necrotic debris, Papanicolaou stain, 4×; (**C**) Abundant necrotic debris, Papanicolaou stain, 20×; (**D**) Detail of the rare histiocytes, Papanicolaou stain, 40×.

**Figure 4 diagnostics-14-02667-f004:**
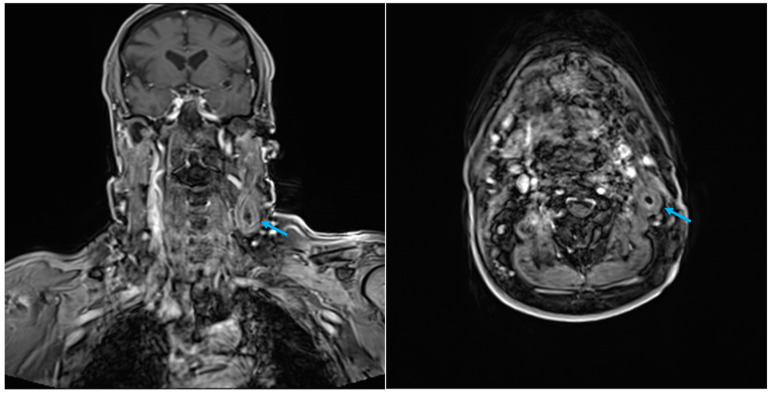
Case 1 MRI, T1 with CE, coronal and axial view. Retained ORC pseudotumour is indicated by the arrow.

**Figure 5 diagnostics-14-02667-f005:**
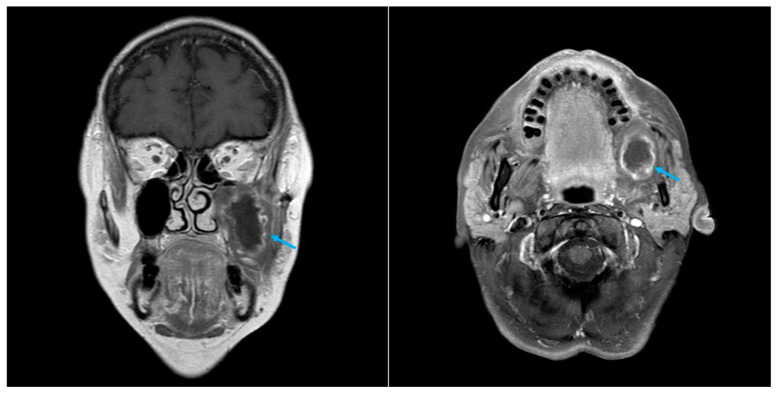
Case 2 MRI, T1 with CE, coronal and axial view. Retained ORC pseudotumour is indicated by the arrow.

**Figure 6 diagnostics-14-02667-f006:**
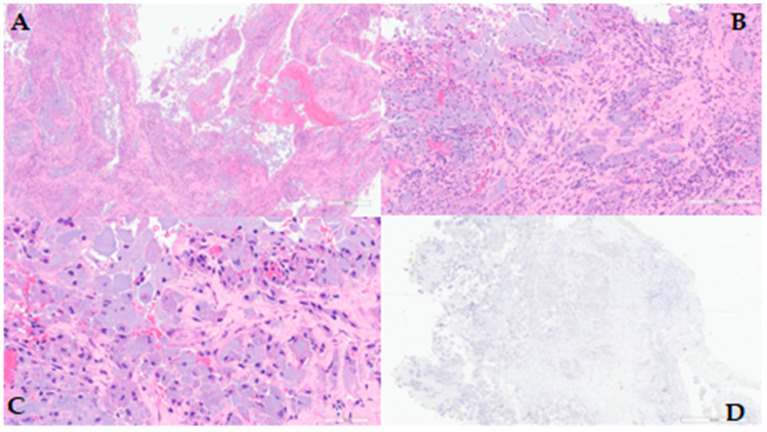
Case 2: (**A**) Fragments from the mass removed, haematoxylin-eosin, 4×; (**B**) Fibrous tissue with chronic inflammation and abundant exogenous material; haematoxylin-eosin, 15×; (**C**) Detail of the histiocytic–macrophagic elements and multinucleated giant cells, haematoxylin-eosin, 40×; (**D**) In immunohistochemistry specimen, 2×.

**Figure 7 diagnostics-14-02667-f007:**
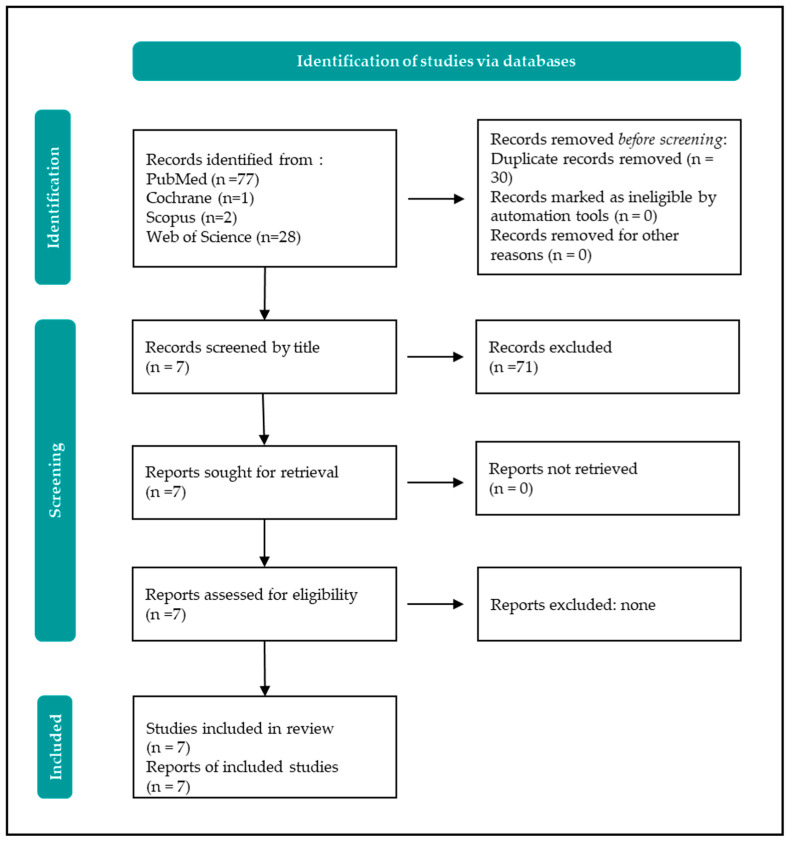
PRISMA 2020 flow diagram for papers selection.

**Table 1 diagnostics-14-02667-t001:** This table shows the PICO method to build the search string. Search string used: (1) AND (2) AND (3).

**Population**	Text word (1): (oxidised cellulose OR surgicel OR tabotamp OR gelita OR absorbable haemostatic) AND (recurrence OR residual OR relapse OR metastasis OR metastatic disease OR abscess OR granuloma OR lymph node OR neck lymphadenopathy)
**Intervention**	Text word (2): (needle aspiration OR surgical removal OR histological examination OR neck dissection OR CT scan OR sonography OR magnetic resonance OR cytological examination OR cytology OR histology)
**Comparison**	Not applicable
**Outcome**	Text word (3): (pseudotumoural lesion OR pseudotumour OR artefact OR granuloma OR (foreign body reaction) OR gossybipoma OR textiloma OR (mimicking of tumour recurrence))

**Table 2 diagnostics-14-02667-t002:** Summary of results. Legend: F = female, M = male, US = ultrasound, CT = computed tomography, FNA = fine-needle aspiration, NA = not available, C = cytology, H = histology, * this number refers to the whole sample in the cited paper, out of the 16 only 4 were considered eligible for this review.

Authors	Year	Study Type	Sample Size	Age	Sex	Surgical Site	Haemostatic Material	Primary Diagnosis	Primary Surgery	Postoperative Imaging	Timing	Diagnostic Strategy	Cytology and/or Histology
**Liu et al.** [12]	2016	Case series	3	36	F	Thyroid	Surgicel	Thyroid carcinoma	right lobectomy and partial left lobectomy	US	24 months 720 d	FNA	C
				52	M	Thyroid	Surgicel	Thyroid adenoma	right lobectomy	US	5 months 150 d	US	NA
				67	F	Thyroid	Surgicel	Thyroid carcinoma	Total Thyroidectomy	US	16 days	US	NA
**Shahin et al.** [13]		Case presentation	1	59	F	Thyroid	Surgicel	Follicular thyroid carcinoma	Two step total thyroidectomy	US	14 days	FNA	C
**Hernandez-Bonilla et al.** [14]	2019	Case series	16 *	36	F	Thyroid	Surgicel	Papillarycarcinoma	Total Thyroidectomy	US	24 months 720 d	FNA	C
				49	F	Thyroid	Surgicel	Multinodular goitre	Total Thyroidectomy	US	13 months 390 d	FNA	C
				35	M	Thyroid	Surgicel	Follicular adenoma	Total Thyroidectomy	US	18 months 540 d	FNA	C
				73	M	Nasopharinx	Surgicel	Nasopharyngeal carcinoma	Excision of primary tutor	NA	15 months 450 d	FNA + cervical lump excision	C + H
**Sebastian-Valles et al.** [15]	2023	Scientific letter	15	mean age 51 (27–77 yo)	F: 10 M: 5	Thyroid	ORC	Differentiated thyroid cancer: 11Non-invasive follicular tumour: 4	Total thyroidectomy: 11Hemithyroidectomy: 4	US	13–27 months mean 20 m 600 (d)	US + C	C
**Liberale et al.** [16]	2023	Case report	1	34	F	Thyroid	Surgicel	Papillary carcinoma	Total thyroidectomy	US + CT + scintigraphy	5 years 1800 d	FNA	C + H
**Azami Hassani et al.** [17]	2021	Case report	1	61	F	ATM	Surgicel	Chondrosarcoma	Excision of primary tumour	MRI	3 months 90d	Surgical excision	H
**Alameer** [18]	2023	Case report	1	42	F	Thyroid	Surgicel	Follicular adenoma	Right thyroidectomy + isthmusectomy	US	2 years 720	FNA + surgical excision	C + H
**Presented case 1**	2024	Case report	1	78	F	Maxilla	Tabotamp	Ameloblastoma	Excision of primary tumour	CT + MRI	15 d	CT + MRI + surgical excison	H
**Presented case 2**	2024	Case report	1	69	M	Oral cavity	Tabotamp	SCC	Hemimandibulectomy, omolateral neck dissection and free flap reconstruction	CT + US	30 d	CT + FNAC	C

**Table 3 diagnostics-14-02667-t003:** Radiological and Histological features of absorbable haemostatics evaluated by ultrasounds (US), Computed Tomography (CT), Magnetic Resonance Imaging (MRI); Positron-Emitted Tomography (PET-CT), Scintigraphy, histology and citology.

Authors	US	CT	MRI	PET/CT	Scintigraphy	Histologic/Cytologic Findings
**Case 1**	Elongated, non-homogenous hyperechogenic formation with an expansive growth pattern	CT (1): Lymphadenopathy […] and exhibited areas of colliquation	Persistent left adenopathy was observed, with no evidence of restricted signal in DWI or post-contrastographic enhancement	Ovoid mass with an elevate 18-FDG uptake (SUV 7, 5)		C (1): Necrotic debris, neutrophilic granulocytes, and rare multinucleated histiocytic giant cells. C (2): Cytological sample revealed the presence of a birefringent material, which was identified as sutures, macrophages, and isolated foreign body-like giant multi-nucleated cells
**Case 2**		Residual tissue with a peripheral solid portion with contrast enhancement and a central hypodense area likely representing fluid, with concurrent air bubbles	Peripheral solid portion, with slight contrast enhancement (T1), and a central colliquated portion			H: Fibrous tissue fragments w histiocytic–macrophagic nature, mixed with neutrophil granulocytes and calcific-like material
**Liu et al.** [12]	Elongated, hypoechoic mass, well-defined margin, no blood flow or calcification, non-uniform punctiform internal echoes					C: Presence of Surgicel fragments
**Shahin et al.** [13]						C: Scattered inflammatory cells (macrophages), amorphous debris
**Hernandez-Bonilla et al.** [14]						C: Variable granulomatous inflammatory reaction w multinucleated giant cells, phagocytic cells. Oxidised cellulose = acellular, laminated inorganic fragments, slender quadrangular appearance.
**Sebastian -Valles et al.** [15]	Hypo-, iso-echogenic, well-defined margins. No doppler flow.					C: Small cellular structures consisting of oxidised cellulose, macrophages.
**Liberale et al.** [16]		Hypodense mass with small intralesional calcification, without post-contrast enhancement			No significant activity	C: Amorphous material H: section of fibrous tissue referable to cystic/pseudocystic wall, with histioid inflammation associated with giant cell reaction from a foreign body
**Azami Hassani et al.** [17]			Well-limited hypointense lesion in T1 and heterogeneous in T2			H: Presence of reworked fibrous material recognised as unabsorbed Surgicel
**Alameer** [18]	Elongated, well-defined isoechoic. No calcifications.					C: Striated muscle fibres, few macrophages, degenerated debris. No epithelial cells. H: Chronic xanthogranolumatous inflammation w foreign body reaction and foreign material. Foamy macrophages, multinucleated giant cells. No atypia.

## Data Availability

There are no restrictions to the data presented in this article.

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
