# Peer review of "Absorbable Haemostatic Artefacts as a Diagnostic Challenge in Postoperative Follow-Up After Oncological Resection in Head and Neck Tumours: Systematic Review of Current Literature and Two Case Reports"

_diagnostics, 2024, doi:10.3390/diagnostics14232667_

Round 1

Reviewer 1 Report

Comments and Suggestions for Authors

The authors reviewed well the literatures about oxidized resorbable cellulose (ORC)-induced pseudotumors in head and neck region. They presented their own two cases with pseudotumor after head and neck surgery. This paper explains how to diagnose ORC)-induced pseudotumors correctly and gives big hints to avoid under- or over-treatment for head and neck cancer patients. All head and neck surgeons should read this article.

Author Response

Comment 1: The authors reviewed well the literatures about oxidized resorbable cellulose (ORC)-induced pseudotumors in head and neck region. They presented their own two cases with pseudotumor after head and neck surgery. This paper explains how to diagnose ORC)-induced pseudotumors correctly and gives big hints to avoid under- or over-treatment for head and neck cancer patients. All head and neck surgeons should read this article.

Reply: Thanks for your comments and positive feedback. Best regards.

Reviewer 2 Report

Comments and Suggestions for Authors

This review is very propitious and seemed to me to be well conducted. There are other consequences of the use of hemostatics. They should not be used indiscriminately, the amount used is also decisive. By the way, this is a suggestion to discuss: does the hemostatic mass used determine these complications?

I believe that the work is ready for publication

Author Response

Comment: This review is very propitious and seemed to me to be well conducted. There are other consequences of the use of hemostatics. They should not be used indiscriminately, the amount used is also decisive. By the way, this is a suggestion to discuss: does the hemostatic mass used determine these complications? I believe that the work is ready for publication.

Reply: Thank you for your comments and your positive feedback. Best regards

Reviewer 3 Report

Comments and Suggestions for Authors

Review for

Absorbable haemostatic artefacts as a diagnostic challenge in postoperative follow up after oncological resection in head and neck tumours: systematic review of current literature and two case-report

  • A brief summary

The article summarizes the imaging and pathological characteristics in pseudotumors generated by the oxidized resorbed cellulose (ORC), using two cases from their departments, but also 26 cases from 7 articles from literature, limited to head and neck surgery. The main contribution of this article is to provide a key for surgeons and radiologists to prevent misinterpretations as recurrency in postoperative oncological cases.

  • General concept comments

The subject of this article is of great interest in postoperative oncological imaging, especially in cases of the head and neck, a difficult region to interpret, which requires experience.

ORCs have been described both from the point of view of their composition and mode of action, but especially from the point of view of the pseudotumor imaging aspects that they can produce when there is not a complete resorption but rather an inflammatory reaction associated with them.

The systematic review followed the guidlines of PRISMA 2020 and the PICO method. The results, following the analysis of the cases in the selected articles, were presented systematized, easy to follow in the form of a table, and the imaging and pathological characteristics that draw attention to a pseudotumor in the case of ORC were subsequently extensively discussed.

The images from the two case report are interpreted appropriately and complete the results.  

The conclusions are clearly noted, systematized on  aspects on each type of imaging investigation (ultrasound,  CT scans with contrast enhancement, MRI, PET-CT, scintigraphy) but but also the histopathological data resulting from FNA or resection.

The references are numerous, including articles published over a long period of time, from 1997 to 2024. There are not included excessive self-citations identified,  and more than half of references have been published in the last five years.

  • Specific comments

I don’t have specific comments.

Author Response

Comment: I don’t have specific comments.

Reply: Thank you for your feedback. Best regards.